# Physical Functional Ability and Quantitative Assessment of the Multifidus Muscle of the Lumbar Spine in the Elderly

**DOI:** 10.3390/diagnostics13142423

**Published:** 2023-07-20

**Authors:** Jung Hae Yun, Dong Gyu Lee

**Affiliations:** Department of Physical Medicine and Rehabilitation, Yeungnam University College of Medicine, Daegu 42415, Republic of Korea; junghae5660@gmail.com

**Keywords:** multifidus, grip strength, elderly, muscle atrophy, fatty infiltration

## Abstract

Aging is associated with muscle atrophy and fatty infiltration of skeletal muscle. The multifidus muscle stabilizes the lumbar spine and undergoes adipose accumulation with age, leading to functional decline in the elderly. Therefore, quantitative assessment of the multifidus muscle can be beneficial for the elderly when formulating treatment strategies and reducing future complications. Fifty-seven patients (mean age, 73.89 ± 6.09; 23 male patients) who underwent lumbar Magnetic resonance imaging (MRI) were prospectively recruited. The cross-sectional area (CSA) of the multifidus from the L2-S1 level and the CSA of the L4-5 level psoas muscle were measured. The functional CSA (fCSA) of the multifidus muscle was measured by excluding the fat infiltration area from the multifidus CSA. The CSA to fCSA ratio was obtained by multiplying 100 by the value obtained by dividing CSA by the fCSA. Pfrrmann classification was used to evaluate the degree of disc degeneration. The functional disability measurements were the Short Physical Performance Battery (SPPB), Berg Balance Scale (BBS), grip strength, and functional reach test (FRT). Pearson’s correlation analysis was used to examine the relationship between the functional disability measurements and the multifidus muscle. The CSA to fCSA ratio value was relatively constant at each spine level and showed a significant correlation with the SPPB, grip strength, FRT, and psoas index (*p* < 0.05). However, degree of disc and multifidus muscle degeneration was not statistically significant. So, age-related changes play a significant role in developing back muscle fatty infiltration than disc degeneration. Moreover, Grip strength showed a stronger relationship with the quality of the multifidus muscle than other functional disability measurements.

## 1. Introduction

Aging causes fatty infiltration of skeletal muscles, which decreases physical performance and quality of life. Sarcopenia, defined as age-related muscle loss and functional decline, is the progressive loss of skeletal muscle mass and strength with advancing age [1]. Diagnosis of sarcopenia was made by quantitative assessment of skeletal muscle mass and quality assessment using the estimation of physical performance [2,3]. Generally, the loss of muscle strength is caused by muscle atrophy and fatty changes, which lead to decreased functional performance. Therefore, the quantitative measurement of muscle mass and fatty infiltration can be a preliminary assessment in the evaluation of sarcopenia.

Proper activation of the core muscle stabilizes the performance of the extremities [4]. Moreover, the core muscles, including the multifidus muscle, stabilize the segmental lumbar spine [5]. In addition, atrophy of the multifidus muscle aggravates further compression of osteoporotic spinal compression fractures [6,7]. Therefore, patients with further atrophy of the back muscle tend to present with more severe back pain than those with less severe back muscle atrophy [8,9]. Hence, core muscle function is essential for maintaining the spine’s stability in older people.

Based on these clinical significances, quantitative assessment of the multifidus muscle can be beneficial for elderly patients to develop a treatment strategy for back pain and to prevent further progression of comorbidities.

The structural and functional decline of skeletal muscles in older people affects the extremities and the axial back core muscles. Moreover, spinal disc degeneration and decreased disc height affect back muscle degeneration [10]. Therefore, our objective was to investigate the impact of multifidus muscle degeneration, specifically its relationship with disc degeneration as a localized factor and with physical performance as a comprehensive measure of overall body changes.

## 2. Materials and Methods

### 2.1. Study Population

Patients who visited our university hospital spine center between August 2021 and October 2022 were prospectively recruited. Patients aged over 60 years, who underwent lumbar spine magnetic resonance imaging (MRI), were able to walk 10 m independently with or without cane support and were capable of functional outcome evaluation were included in the study. Patients with a history of lumbar spine surgery, severe degenerative scoliosis, spinal infection, communication disorders, diseases that severely limit physical activity, including encephalopathy or severe musculoskeletal disorders, and who were unable to perform functional evaluation due to pain were excluded. This study was reviewed and approved by the Institutional Review of our hospital (OOOO 2021-08-053). All Subjects included in this study provided written informed consent.

### 2.2. Multifidus Measurement from Spine MRI

Lumbar spine magnetic resonance imaging was performed using a 3.0-T magnetic resonance unit (Siemens, Munich, Germany). Axial T2-weighted MRI (repetition time: 3400–3700 ms, echo time: 90–111 ms) was used to measure the cross-sectional area (CSA) and fatty infiltration of the multifidus muscles at the level of the L2-S1 intervertebral disc. We used image-processing software (version 2.9.0 ImageJ; National Institutes of Health, Bethesda, MD, USA) to measure the CSA and fatty infiltration of the multifidus muscles. DICOM files extracted from lumbar spine MRI were mounted using the ImageJ program. The CSA of the multifidus muscle was measured by manually outlining the border of each muscle. The fatty infiltration area was measured using the pseudo-coloring technique by converting the 32-bit image to a pseudo-color 8-bit image, followed by threshold analysis, in which the optimal threshold of fat tissue was set manually. Fatty infiltration on axial T2-weighted MR images was converted to red color. The red proportion of CSA of the multifidus muscle was calculated. Functional muscle mass excluding fat tissue, defined as functional CSA (fCSA), was calculated by subtracting the fat area from the total multifidus CSA (Figure 1). The ratio between the total CSA and fCSA was calculated by dividing the fCSA by the total CSA. In this study, we presented the CSA to fCSA ratio as a percentage by multiplying 100 with the value obtained by dividing CSA by fCSA. In this study, we presented the CSA to fCSA ratio as a percentage by multiplying 100 with the value obtained by dividing CSA by fCSA.

### 2.3. Pfirrmann Grading System

The Pfirrmann grading system is a widely used classification system for intervertebral disc degeneration, which grades disc degeneration from I to V using T2 weighted MR images. Disc degeneration is classified according to the signal intensity, distinction between the annulus and nucleus, and height of the intervertebral disc [11]. Grade I is a condition in which the disc structure is homogeneous and has a bright and hyperintense white signal intensity with normal disc height. Grade II occurs when the structure of the disc remains hyperintense, with distinct annulus and nucleus margins, and normal height, but the structure is inhomogeneous. Grade III has an inhomogeneous disc structure with intermediate gray signal intensity and unclear nucleus and annulus distinction; however, the disc height is normal or slightly decreased. Grade IV has an inhomogeneous disc structure with hypointense dark gray signal intensity, no distinction between the nucleus and annulus, and a normal or moderately decreased disc height. Grade V is classified when the disc is inhomogeneous, has a hypointense black signal intensity, no distinction between the nucleus and annulus, and has a collapsed disc space. We used the Pfirrmann classification to classify the degree of degeneration of the intervertebral disc of L2-S1.

### 2.4. Measures

The Short Physical Performance Battery (SPPB) is a commonly used tool for testing physical performance in elderly populations, with high reliability and validity [12]. The SPPB test consists of three subtests: standing balance, time required to walk 4 m, and time taken to perform five sit-to-stand exercises on a chair consecutively. For balance testing, the participants were asked to stand for at least 10 s, if possible, in three positions. First, by positioning their feet as close as possible, then in a semi-tandem posture, in which one foot is placed behind the other, and in a tandem posture, in which one foot is postured directly behind the other touching it. For the first two positions, 0 or 1 point was given depending on whether it could be maintained for more than 10 s. For the last position, one point was given if the participant was able to hold for 3–10 s and two points if it could be maintained for more than 10 s. The total score is calculated as the sum of the three positions. As a result, a score of at least 0–4 was given for balance testing. For the 4 m walking test, the time required to walk 4 m at the usual pace was measured. The time to stand up from a chair five times consecutively was measured as quickly as possible. Scores of 1–4 based on the time spent were measured for the last two of the three tests. A total score of 0–12 was obtained as the sum of all three tests.

The Berg Balance Scale (BBS) is a well-established and valid method for predicting static and dynamic activities of everyday living [13]. The test included 14 static and dynamic activities of daily living, including sit-to-stand, independent standing, tandem stepping, and one-leg standing. Each item was evaluated on a scale of 0–4, depending on the degree of function, with a total score of 56 points. A higher score indicates higher functional ability.

Grip strength is a widely used measure of geriatric conditions including frailty. Although grip strength measures the strength of hand muscles, it has been widely used in the elderly to assess the generalized effect of the musculoskeletal system in clinical practice [14]. The Southampton protocol was used for the consistency of hand grip measurement, in which hand grip was measured using the hand dynamometer alternately from right to left hand three times in total, with the forearm comfortably placed on the chair arm in a sitting position [15]. Subsequently, the average strength of the measurements was determined.

The functional reach test (FRT) measures the difference between the maximal forward reach and the patient’s arm length in an independent standing posture with the arm elevated to shoulder height. FRT is a precise, inexpensive, and reliable method to measure postural stability [16,17]. This study aimed to predict postural instability during daily activities using the FRT.

Finally, the psoas index, which is a widely used index for predicting sarcopenia, was measured. The Psoas index is correlated with back muscle degeneration [18]. The psoas muscle index was obtained by dividing the bilateral psoas muscle CSA at the L4-5 level by the square of the height.

### 2.5. Statistical Analysis

Statistical analysis was performed using SPSS version 22.0 (SPSS Inc., Chicago, IL, USA), and continuous variables were presented as mean ± standard deviation. ANOVA analysis was used to evaluate whether differences existed between lumbar spine levels of total multifidus CSA, fCSA, and the CSA to fCSA ratio. Tukey’s Honest Significant Difference (HSD) test was conducted as a post hoc analysis to assess significant differences among lumbar spine levels within each group. In addition, one-way analysis of variance was used to evaluate the CSA of the multifidus, fCSA, and CSA to fCSA ratio, classified according to the Pfirrmann Classification of lumbar discs at each level. Finally, Pearson’s correlation analysis was used to evaluate how each lumbar spine level multifidus parameter was correlated with functional outcome measurements (SPPB, BBS, Grip strength, and FRT) and psoas index. Statistical significance was set at *p* < 0.05.

## 3. Results

A total of fifty-seven patients (mean age 73.89 ± 6.09; 23 male patients) visiting our hospital were prospectively recruited. The demographic characteristics of the participants are presented in Table 1.

The CSA of the multifidus, fCSA, and CSA to fCSA ratio are presented in Table 2 and Figure 2. The CSA of the multifidus, fCSA, and CSA to fCSA ratio showed statistically significant differences between lumbar spine levels within each group (*p* < 0.001, *p* < 0.001, and *p* = 0.022, respectively). However, in the case of the CSA to fCSA ratio values, it was confirmed that there was a relatively constant value among the spine levels when examined through post hoc analysis.

Using one-way ANOVA analysis, the relationship between the degree of disc degeneration and the multifidus, fCSA, and CSA to fCSA ratio were investigated (Appendix A). The degree of disc degeneration and CSA of the multifidus and fCSA and the CSA to fCSA ratio were not statistically significant. However, the CSA to fCSA ratio of the L5-S1 level showed significantly higher fatty infiltration according to the degree of disc degeneration (*p* < 0.001). L5-S1 level was more affected by disc degeneration than the other spinal levels.

Finally, the correlation between the physical functional measurements and multifidus CSA, fCSA, and the CSA to fCSA ratio was analyzed using Pearson correlation analysis. Moreover, the correlation between the psoas index and multifidus parameters was analyzed (Table 3). The psoas index demonstrated statistically significant correlations with all parameters of the multifidus muscles. Grip strength and the psoas index showed moderate to strong significant correlations with each factor concerning the multifidus muscles at all levels of the lumbar spine. Moreover, all physical functional tests, except BBS, showed a significant correlation with the CSA to fCSA ratio.

## 4. Discussion

The multifidus muscle strongly correlated with the psoas index and functional performance measurements. Grip strength had a more significant relationship than other measurements. The CSA to fCSA Ratio was evenly distributed on each spinal level compared to the CSA and fCSA of the multifidus muscle. Disc degeneration and the Pfirrmann Classification did not affect multifidus degeneration except at the L5-S1 level. We speculated that multivariant factors, including disc degeneration, affected the multifidus fatty infiltration at the L5-S1 level. We speculated that in older adults, the aging process exerts a more substantial influence on the fatty infiltration of back muscles compared to disc degeneration.

Atrophy and functional decline of the skeletal muscles are closely related to aging. Muscle decline that occurs with aging is associated with the accumulation of adipose tissue. Unloading, disuse, and sex steroid deficiency, such as estrogen deficiency in women and androgen deprivation therapy in men, are associated with increased adipogenesis in the elderly. Adipogenesis is known to lower the sensitivity of skeletal muscles to insulin. As insulin acts as a skeletal anabolic factor [19], the vicious cycle that ensues leads to muscle atrophy. Therefore, aging and adipogenesis lead to the deterioration of muscle mass, strength, and physical function.

Core muscles, including the multifidus, play an important role in back stabilization. Therefore, atrophy and fatty infiltration of these muscles are associated with back pain and disc degeneration [20,21]. In addition, during degeneration, the back muscles, including the multifidus, undergo histological changes. The elderly have a higher proportion of type II muscle fibers compared to healthy individuals [22] and have transformed from oxidative to glycolytic fibers [22,23]. Type 2 and glycolytic fibers are susceptible to fatigue, which may lead to decreased endurance and functional disability. In previous studies, exercise-induced co-contraction of the deep trunk muscles, including the multifidus muscle, reduced back pain [24,25]. Studies have investigated the relationship between multifidus muscle and pain in patients with low back pain. These studies provide evidence of a strong correlation between back pain and multifidus muscle. So, our study aimed to examine the relationship between physical performance and multifidus degeneration. Therefore, proper exercise prescription for elderly patients is important, based on objective and quantitative evaluation of the back muscles.

Previous studies claim that disc degeneration is related to core muscle degeneration [10,11]. Intervertebral discs are responsible for two-thirds of axial spinal loading. Also, radiculopathy caused by disc herniation can lead to multifidus atrophy. However, another study showed no relationship between trunk muscle mass and pain intensity [26]. Compared to discectomy, spinal fusion surgery results in a significant decrease in multifidus muscle [27]. However, both surgical procedures show similar levels of pain reduction for recurrent lumbar disc herniation [28]. Therefore, the relationship between multifidus atrophy, back pain, and disc degeneration remains controversial. In this study, we found little correlation between disc and multifidus muscle degeneration. Only the CSA to fCSA ratio of L5-S1 level was statistically significant (*p* < 0.001). Therefore, researchers should consider the effect of age or sarcopenia on back muscle atrophy in addition to disc degeneration.

In the present study, quantitative measurement of the CSA and fCSA of multifidus muscle for each level between L2-S1 increased toward the caudal side. This result is thought to be due to anatomical characteristics. However, in the case of the CSA to fCSA ratio, each level showed a relatively constant value. Moreover, the CSA to fCSA ratio and Psoas index showed a significant relationship. Functional disability measurements showed a significant relationship with this CSA to fCSA ratio. As a result, the CSA to fCSA ratio index was related to the structural and functional measurements of sarcopenia. Therefore, the CSA to fCSA ratio of the back muscles can be a potential index for sarcopenia estimation.

The SPPB, BBS, Grip strength, and FRT were used to measure functional disability in this study. The SPPB is an objective index for measuring daily life performance, including balance and gait, with high reliability and validity [12]. The BBS is a method for measuring static and dynamic balance related to activities of daily living. The BBS is an indicator that specifically reflects the balance among functional abilities. The FRT was used to measure postural stability. Grip strength was used as an index to represent the musculoskeletal system of the entire body to measure the overall geriatric condition [14]. These functional disability measurements showed relatively consistent correlations at each level of the multifidus muscle parameters. However, no significant correlation was observed between BBS scores. This may be because the BBS evaluates balance itself intensively rather than the overall functional disability, and balance can be affected by the integration of various factors, including proprioceptive, vestibular, and visual sensory systems [29,30,31].

Grip strength strongly correlated with all parameters of the multifidus muscle. Several previous studies have used grip strength as a predictor of functional disability in the elderly. Therefore, grip strength is commonly used as a representative measurement of overall geriatric condition [14,32]. In addition to the elderly, grip strength in middle-aged individuals has also been used as a predictor of disability in old age. In other words, grip strength is a cause and result of geriatric conditions [33,34], while other measurements reflect the consequential consequences of geriatric conditions. Therefore, grip strength reflects geriatric condition better than other measurements. Hence, grip strength may be an optimal functional evaluation parameter for back muscle degeneration.

However, this study had several limitations. First, a relatively small number of patients were recruited. Second, we analyzed the multifidus muscle among the various back muscles involved in spinal stabilization. However, the multifidus plays a crucial role in spinal segmental stabilization among the back muscles [35]. Therefore, we believe that the analysis of the multifidus muscle holds significant clinical relevance.

## 5. Conclusions

The ratio of fatty infiltration in the multifidus muscle showed a relatively constant value for each spinal level and had a significant and higher correlation with the psoas index and functional disability measures, including SPPB, FRT, and grip strength. Therefore, in elderly patients, back muscle fatty infiltration is significantly influenced by aging rather than disc degeneration. Thus, evaluation and treatment for back muscle atrophy and fatty infiltration can be considered within the same aging category, similar to the decline in muscle mass that occurs in older individuals. Moreover, grip strength showed a high correlation with all parameters of the multifidus muscle. As a result, grip strength can be used as an optimal functional evaluation parameter in studies related to back muscle degeneration.

## Figures and Tables

**Figure 1 diagnostics-13-02423-f001:**
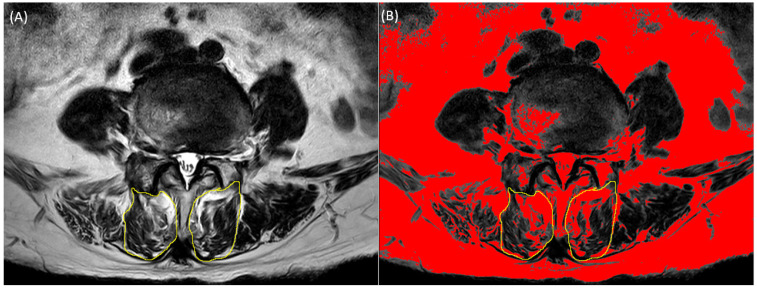
Manually traced region of multifidus muscle. (**A**) Multifidus muscle (yellow circle) traced on Axial T2-weighted MR image. (**B**) Multifidus muscle traced on converted axial T2-weighted MR image using pseudo-coloring technique in which fatty infiltration is colored red.

**Figure 2 diagnostics-13-02423-f002:**
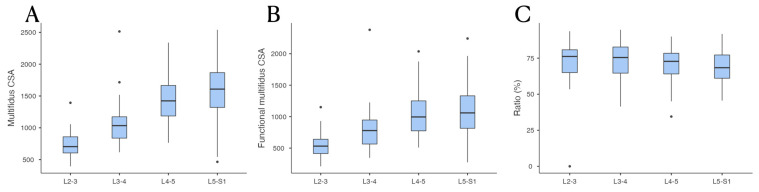
Comparisons of the mean value between spinal levels among groups. (**A**) Comparison of CSA of multifidus muscle. (**B**) Comparison of CSA of functional multifidus muscle. (**C**) Comparison of the CSA to fCSA ratio.

**Table 1 diagnostics-13-02423-t001:** Baseline demographics.

Characteristics	*N* = 57
Age (y)	73.89 (6.09)
Gender (*n*, % male)	23 (40.35)
Height (kg)	1.57 (0.08)
Weight (m)	60.03 (9.56)
BMI (kg/m^2^)	24.21 (2.98)
SPPB	9.39 (1.93)
BBS	52.42 (2.98)
Grip (kg)	24.03 (7.06)
FRT (cm)	19.56 (7.22)
Psoas index (cm^2^/m^2^)	7.80 (2.17)

Values are presented as mean (standard deviation) unless otherwise stated. Abbreviations: BMI, body mass index; SPPB, short physical performance battery; BBS, Berg Balance Scale; FRT, Functional Reach Test. Psoas index: bilateral psoas muscle CSA/height^2^.

**Table 2 diagnostics-13-02423-t002:** CSA of multifidus and functional multifidus muscle and the CSA to fCSA ratio.

	Multifidus (cm^2^)		Functional Multifidus (cm^2^)		Ratio (%)	
Group	M ± SD	F (*p*-Value)	M ± SD	F (*p*-Value)	M ± SD	F (*p*-Value)
L2-3	727.58 ± 184.28 ^a^	66.141	546.04 ± 179.91 ^a^	30.112	74.17 ± 10.85 ^b^	3.287
L3-4	1058.75 ± 301.98 ^b^	(<0.001)	793.58 ± 317.03 ^b^	(<0.001)	73.58 ± 11.58 ^ab^	(0.022)
L4-5	1455.5 ± 360.06 ^c^		1038.02 ± 358.49 ^c^		70.54 ± 11.39 ^ab^	
L5-S1	1550.94 ± 484.79 ^c^		1079.81 ± 440.78 ^c^		68.27 ± 11.83 ^a^	
Total	1200.26 ± 478.62		865.77 ± 398.06		71.63 ± 11.60	

ANOVA Post hoc analysis: Tukey, Groups marked with the same superscript letter (a–c) indicate that they are not significantly different from each other according to the post hoc test. Values are presented as mean ± standard deviation. Ratio (%): CSA of multifidus muscle/CSA of functional multifidus muscle × 100. Abbreviations: CSA, cross-sectional area.

**Table 3 diagnostics-13-02423-t003:** Correlation between CSA and fCSA of multifidus muscle and the CSA to fCSA ratio of each level with functional outcomes and Psoas index using Pearson’s correlation coefficient analysis.

	L2-3		L3-4		L4-5		L5-S1	
Multifidus CSA	r	*p*-Value	r	*p*-Value	r	*p*-Value	r	*p*-Value
SPPB	0.048	0.727	0.144	0.284	0.213	0.112	0.065	0.632
BBS	0.177	0.077	0.179	0.182	0.275	0.039 *	0.061	0.654
Grip	0.405	0.002 *	0.583	<0.001 *	0.588	<0.001 *	0.393	0.003 *
FRT	0.119	0.119	0.273	0.040 *	0.181	0.177	0.228	0.088
Psoas index	0.383	0.004 *	0.406	0.002 *	0.454	<0.001 *	0.285	0.032 *
**Functional CSA**	**r**	** *p* **	**r**	** *p* **	**r**	** *p* **	**r**	** *p* **
SPPB	0.186	0.169	0.269	0.043 *	0.300	0.024 *	0.198	0.139
BBS	0.138	0.138	0.238	0.074	0.290	0.028 *	0.141	0.295
Grip	0.572	<0.001 *	0.687	<0.001 *	0.612	<0.001 *	0.520	<0.001 *
FRT	0.254	0.059	0.311	0.019 *	0.325	0.014 *	0.342	0.009 *
Psoas index	0.491	<0.001 *	0.463	<0.001 *	0.471	<0.001 *	0.296	0.026 *
**Ratio**	**r**	** *p* **	**r**	** *p* **	**r**	** *p* **	**r**	** *p* **
SPPB	0.284	0.034 *	0.367	0.005 *	0.339	0.010 *	0.362	0.006 *
BBS	0.143	0.292	0.218	0.103	0.213	0.111	0.213	0.057
Grip	0.530	<0.001 *	0.580	<0.001 *	0.331	0.012 *	0.331	<0.001 *
FRT	0.366	0.006 *	0.278	0.036 *	0.373	0.004 *	0.373	0.002 *
Psoas index	0.392	0.003 *	0.393	0.002 *	0.271	0.042 *	0.409	0.002 *

Ratio (%): CSA of multifidus muscle/CSA of functional multifidus muscle × 100. Abbreviations: CSA, cross-sectional area; SPPB, short physical performance battery; BBS, Berg Balance Scale; FRT, Functional Reach Test. Psoas index: bilateral psoas muscle CSA/height^2^. * *p* < 0.05.

## Data Availability

Data are available on a reasonable request from the authors.

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
