# Peer review of "Physical Functional Ability and Quantitative Assessment of the Multifidus Muscle of the Lumbar Spine in the Elderly"

_diagnostics, 2023, doi:10.3390/diagnostics13142423_

Round 1

Reviewer 1 Report

This topic is interesting, but several changes are needed. Look at these:

- Lines 52-54: "Therefore, we estimated the relationship between disc degeneration, physical performance, and quantitative decline of the multifidus muscles at the segmental level of the lumbar spine" It is not clear what is the purpose of this paper. Improve this part.

- Lines 67-68: Figure 1 is not present in the text. In addiction, what's the point of this photo? what kind of technique is this?

- Lines 187-188: "Moreover, the correlation between the psoas index and multifidus parameters was analyzed (Table 3)" Describe results in the text.

-Lines 231-239: "Therefore, the relationship between multifidus atrophy, back pain, and disc degeneration remains controversial. In this study..." Authors must consider these very important papers: ---- Ahsan et al. Fusion versus nonfusion treatment for recurrent lumbar disc herniation. J Craniovertebr Junction Spine. 2021  ----  Lewandrowski et al. The Changing Environment in Postgraduate Education in Orthopedic Surgery and Neurosurgery and Its Impact on Technology-Driven Targeted Interventional and Surgical Pain Management: Perspectives from Europe, Latin America, Asia, and The United States. J Pers Med. 2023    ----  Nurmukhametov et al. Transforaminal Fusion Using Physiologically Integrated Titanium Cages with a Novel Design in Patients with Degenerative Spinal Disorders: A Pilot Study. Surgeries 2022.

- Lines 273-274: "Exercise programs can induce reversible changes in the CSA and reduce pain... " Improve this point.

- Lines 283-290. "Conclusion" What is the author's opinion on their results?

Minor editing of English language required.

Author Response

We appreciate the insightful and helpful comments of the editor and reviewers very much. We have made as many changes as possible according to the reviewer’s recommendations and have prepared the responses in a point-by-point fashion. We hope that our revision is satisfactory to the standards of the editors and reviewers and look forward to hearing the ultimate decision.

- Lines 52-54: "Therefore, we estimated the relationship between disc degeneration, physical performance, and quantitative decline of the multifidus muscles at the segmental level of the lumbar spine" It is not clear what is the purpose of this paper. Improve this part.

Author’s response: To clarify the purpose of the study, the following revision has been made: "Therefore, our objective was to investigate the impact of multifidus muscle degeneration, specifically its relationship with disc degeneration as a localized factor and with physical performance as a comprehensive measure of overall body changes.."

- Lines 67-68: Figure 1 is not present in the text. In addiction, what's the point of this photo? what kind of technique is this?

Author’s response: Figure 1 is in the main text at line 86. It depicts the method for measuring the multifidus muscle described in the Methods section. The references for the image analysis using FIJI is provided below:

  1. Long, Douglas E., et al. "A guide for using NIH Image J for single slice cross-sectional area and composition analysis of the thigh from computed tomography." PloS one2 (2019): e0211629.
  2. Lee, Dong Gyu, and Jae Hwa Bae. "Fatty infiltration of the multifidus muscle independently increases osteoporotic vertebral compression fracture risk." BMC Musculoskeletal Disorders1 (2023): 1-8.

- Lines 187-188: "Moreover, the correlation between the psoas index and multifidus parameters was analyzed (Table 3)" Describe results in the text.

Author’s response: We inserted the following comment in line 189: "The psoas index demonstrated statistically significant correlations with all parameters of the multifidus muscles."

-Lines 231-239: "Therefore, the relationship between multifidus atrophy, back pain, and disc degeneration remains controversial. In this study..." Authors must consider these very important papers: ---- Ahsan et al. Fusion versus nonfusion treatment for recurrent lumbar disc herniation. J Craniovertebr Junction Spine. 2021  ---

Author’s response: We have included a recommended and significant study to investigate the relationship between the multifidus muscle and back pain.

-  Lewandrowski et al. The Changing Environment in Postgraduate Education in Orthopedic Surgery and Neurosurgery and Its Impact on Technology-Driven Targeted Interventional and Surgical Pain Management: Perspectives from Europe, Latin America, Asia, and The United States. J Pers Med. 2023    ---

-  Nurmukhametov et al. Transforaminal Fusion Using Physiologically Integrated Titanium Cages with a Novel Design in Patients with Degenerative Spinal Disorders: A Pilot Study. Surgeries 2022.

Author’s response: Thank you for the recommendations and suggestions regarding critical-related studies. While we appreciate the suggested research, it seems that the nature of our study may not align directly with those studies, making it challenging to incorporate them into our research or modify the content accordingly. Nonetheless, we genuinely appreciate your valuable input, and we will certainly consider the broader context of the recommended studies when discussing the implications and future directions of our research. Thank you for understanding the constraints we face in this situation.

- Lines 273-274: "Exercise programs can induce reversible changes in the CSA and reduce pain... " Improve this point.

Author’s response: The mentioned section emphasizes the critical role of multifidus in back pain and the potential for improving muscle strength through appropriate treatment, which can lead to a reduction in pain. While it is necessary to provide additional details, as you mentioned, the submission guidelines impose restrictions on the word count. Therefore, we added further descriptions regarding multifidus at line 232, where relevant content is included.

- Lines 283-290. "Conclusion" What is the author's opinion on their results?

Author’s response: The authors' opinions in this study are twofold. Firstly, the multifidus muscle demonstrates an anatomical characteristic where the cross-sectional area widens as it descends caudally along the spinal levels. Hence, analyzing the simple cross-sectional area alone poses challenges in assessing its correlation with physical performance. To address this, we aimed to quantify this anatomical characteristic by analyzing fatty infiltration and exploring its quantitative relationship with physical performance. Therefore, through this study, we believe it is possible to analyze spinal levels using the fatty infiltration ratio of the multifidus.

Secondly, we examine the correlation between each spinal level and physical performance measurements. Disc pathology resulting from disc degeneration exhibits a higher prevalence in the lower lumbar levels. Therefore, we sought to investigate whether the quantitative results of multifidus degeneration vary according to the spinal level. The fatty infiltration ratio of the multifidus occurs relatively evenly across the spinal levels.

Reviewer 2 Report

The authors have found an interesting observation associated between multifidus muscle and aging. Fatty infiltration in multifidus muscles can be detrimental leading to issues related to mobility and strength. Therefore the observation made by these authors can be considered as an important finding. The authors have assessed 57 male individuals in their study.  It would have been more informative by including female individuals and a more generalized conclusion could have been derived. Overall I found the study to be interesting and informative. Here are my comments.

1.       The authors have used samples from male individuals for their study. Depending on factors like sex, comorbidities, or levels of physical activity, performing a subgroup analysis could have provided more information. This can help in evaluating potential differences in the relationships between the characteristics of multifidus muscles and functional impairment in different subpopulations.

2.       Have the authors examined and considered any additional factors, such as hormonal changes, food, levels of physical activity, or medications that may have an impact on muscle atrophy and fatty infiltration. By taking into consideration these factors, it is feasible to improve the results' dependability and learn more about how aging influences muscle changes.

3.       The authors assessed the data from 57 individuals. Expanding the number of participants can improve the statistical significance and generalizability of the findings. A larger sample size can produce more representative data, which can enhance the reliability of the conclusions drawn.

4.       Including a longitudinal study that follows people over time could have provided additional info. This would allow researchers to study the aging of the multifidus muscle as well as the progression of fatty infiltration and muscular atrophy. Longitudinal data can provide stronger evidence of causal relationships.

5.       Please consider including a control group of young individuals who do not experience muscular atrophy due to aging. By comparing the multifidus muscle characteristics between the elderly group and a control group, it would be possible to more clearly identify how aging impacts muscle atrophy and fatty infiltration.

6.       In addition to Pearson's correlation analysis, the authors may include multivariate analytic techniques like regression analysis or structural equation modeling. These methods can be used to ascertain both the interrelationships between different parameters (such age, the CSA to fCSA ratio, and disc degeneration) and their individual contributions to functional impairment.

7. The labelling in the graphs in Figure 2 can be improved by using a larger font size.

Round 2

Reviewer 1 Report

Authors solved all my criticisms.

 Minor editing of English language required